# Optimal Classification under Performative Distribution Shift

**Edwige Cyffers**
Univ. Lille, Inria, CNRS, Centrale Lille,
UMR 9189 - CRIStAL, F-59000 Lille
`edwige.cyffers@inria.fr`

**Muni Sreenivas Pydi**
Université Paris Dauphine, Université PSL,
CNRS, LAMSADE, 75016 Paris

**Jamal Atif**
Université Paris Dauphine, Université PSL,
CNRS, LAMSADE, 75016 Paris

**Oliver Cappé**
École Normale Supérieure, Université PSL,
CNRS, Inria, DI ENS, 75005 Paris

## Abstract

Performative learning addresses the increasingly pervasive situations in which algorithmic decisions may induce changes in the data distribution as a consequence of their public deployment. We propose a novel view in which these performative effects are modelled as push-forward measures. This general framework encompasses existing models and enables novel performative gradient estimation methods, leading to more efficient and scalable learning strategies. For distribution shifts, unlike previous models which require full specification of the data distribution, we only assume knowledge of the shift operator that represents the performative changes. This approach can also be integrated into various change-of-variable-based models, such as VAEs or normalizing flows. Focusing on classification with a linear-in-parameters performative effect, we prove the convexity of the performative risk under a new set of assumptions. Notably, we do not limit the strength of performative effects but rather their direction, requiring only that classification becomes harder when deploying more accurate models. In this case, we also establish a connection with adversarially robust classification by reformulating the minimization of the performative risk as a min-max variational problem. Finally, we illustrate our approach on synthetic and real datasets.

## 1 Introduction

Machine learning models are increasingly deployed in real-world scenarios where their predictions can influence the users' behaviors, thereby altering the underlying data distribution. This phenomenon, though rooted in long-standing economic theory [Morgenstern, 1928, Muth, 1961], has recently attracted interest in the machine learning community under the name of *performative prediction* [Perdomo et al., 2020, Hardt and Mendler-Dünner, 2023]. Consider for instance a social ranking system: if it consistently favors a particular subpopulation of individuals, user behavior might shift towards mimicking the main characteristics of this subgroup or, conversely, some features of this subpopulation can undergo modification as a consequence of the selection by the system, both effects leading to subtle alterations of the original data distribution. More generally, performative learning captures dynamics at stake in strategic classification, where individuals are confronted by algorithmic decisions that impact their life – such as loan acceptance, college admission, probation – and might thus try to overturn predictions by optimizing some of their features.

This feedback loop, where predictions influence future data, poses new challenges and necessitates the development of novel approaches within statistical learning theory and practice [Perdomo et al.,

2020, Jagadeesan et al., 2022, Drusvyatskiy and Xiao, 2023, Hardt and Mendler-Dünner, 2023, Zezulka and Genin, 2023]. Perdomo et al. [2020] proposed to formalize performative learning as a generalized risk minimization problem, with the *performative risk* being defined as

$$\mathrm{PR}(\theta) = \mathbb{E}_\theta[\ell(Z; \theta)], \tag{1}$$

where $\ell$ is a loss function, $\theta$ a model's parameters, and $Z$ an observable random variable drawn from a distribution $\mathbb{P}_\theta$ also parametrized by $\theta$ itself. In light of the difficulty of minimizing $\mathrm{PR}(\theta)$ directly, one can define a *decoupled performative risk* as $\mathrm{DPR}(\theta, \theta') = \mathbb{E}_\theta[\ell(Z; \theta')]$, clarifying the interplay between the model's prediction and the distribution change. This can be seen as a Stackelberg game that stabilizes when neither the modeler (learned parameters) nor the environment (distribution) has incentive to change their states. Solving the performative learning problem consists in minimizing this risk under the constraint that $\theta = \theta'$, because the testing samples will follow the distribution corresponding to the deployed model, and thus $\mathrm{PR}(\theta) = \mathrm{DPR}(\theta, \theta)$. Minimizing $\mathrm{DPR}(\theta, \theta')$ w.r.t. $\theta'$ for a fixed $\theta$ corresponds to the classical machine learning setting. In contrast, estimating the performative effect, i.e., knowing how to optimize $\theta$ for a given $\theta'$ is more challenging as, per definition, one can only perform statistics from samples collected for values of the parameters $\theta$ for which the model has already been deployed. Hence, performative learning does require some form of counterfactual extrapolation, i.e., what will happen to the data distribution when the parameter $\theta$ changes from its current setting?

Hence, instead of focusing on methods finding performatively optimal points, $\theta_{PO} \in \arg\min \mathrm{PR}(\theta)$, many previous works, following Perdomo et al. [2020], focus on finding stable points $\theta_{PS} \in \arg\min \mathrm{DPR}(\theta_{PS}, \theta)$, through methods that iteratively minimize the empirical risk. This line of research is appropriate in settings where the performative effect can be tamed. If it is sufficiently small, explicitly taking into account the performative changes of distribution is not required and optimal and stable points will be close enough [Perdomo et al., 2020]. However, real use cases do not always satisfy such strong assumptions (see further discussion in Section 4). In general, stable points may not be good proxys for performatively optimal points, particularly in settings where the performative effect cannot be bounded a priori.

Towards this goal, another line of research focuses on finding the optimal points $\theta_{PO}$. Izzo et al. [2022] propose to use Monte Carlo sample-based approximations of the gradient of the performative risk, $\nabla_\theta \mathrm{PR}(\theta)$, based on the score function estimator (see Section 2 below). Miller et al. [2021] use a two-stage approach that deploys random models to estimate the performative effect in the first stage, and then minimizes the estimated performative risk in the second stage. A drawback of both of these approaches is the restrictive set of assumptions needed to show that the algorithms converge. While Izzo et al. [2022] assume the convexity of $\mathrm{PR}(\theta)$ along with smoothness and boundedness contitions, Miller et al. [2021] assume that the loss function is simultaneously strongly convex and smooth. Moreover, the score function estimator of Izzo et al. [2022] necessitates full knowledge of a parametric form of $\mathbb{P}_\theta$, which is unrealistic in practice. Alternatively, Jagadeesan et al. [2022] resort to derivative-free (or zeroth-order) optimization strategies. However, such an approach is appropriate only when it is possible to sequentially deploy a large number of model instances, and it does not scale with the dimension of the parameter $\theta$.

The present work is connected to the second line of the research explored above, where the focus is on finding the optimal point $\theta_{PO}$. Our contributions are as follows.

**(i) Model the performative effect as a push-forward operator** This novel approach provides a *new explicit expression of the performative gradient*. Not only does this approach allow estimation of the performative gradient in settings where previous methods couldn't, but we show that in typical use cases, the variance of this new estimator is significantly smaller.

**(ii) Convexity for Performative Classification** We then focus on the specific task of strategic classi­fication, as this performative learning problem encompasses various real-use cases with important societal impact such as college admission or credit decisions. Our second contribution is to provide new convexity results on the performative risk in this case. Whereas existing results were only proving convexity under assumptions restricting the performative effect to be small compared to the (assumed) strong convexity of the loss function $\ell(z; \theta)$, our results leverage structural assumptions on the performative effect that ensure that the performative risk is convex *without any restriction on the strength of the performative effect*.

**(iii) Linking Performative and Robust Learning** We establish a connexion between performative learning and adversarially robust learning, paving the way to transferring robustness results to the

performative learning field. In particular, this result gives new insights on the empirical evidence in favor of using regularization in the presence of performative effets. Finally, we illustrate our findings on synthetic and real-world datasets.

## 2 Push-forward Model for Performative Effects

In this section, we study the general performative learning setting without yet specializing it to the classification context. In Section 2.1, we introduce the push-forward model of performative learning and derive the expression of the gradient of the performative risk under this model. In Section 2.2, we present a reparameterization-based estimator for the gradient of the performative risk, and compare it to the score function based estimator considered by Izzo et al. [2022].

### 2.1 The Push-forward Model

We aim to minimize the performative risk defined in eq. (1), where the observation $Z$ is drawn from the distribution $\mathbb{P}_\theta$, which depends on the parameter $\theta \in \mathbb{R}^p$ of the learning model. For this to be tractable, one needs additional hypotheses on the nature of the performative effect. We propose to represent the performative effect through a push-forward measure, which matches the intuition of having an untouched distribution that is steered by the performative effect.

**Assumption 1** (Push-forward Performative Model). *For a given model parameter $\theta \in \mathbb{R}^p$, the samples' distribution under the performative effect is given by $\mathbb{P}_\theta = \varphi(\cdot; \theta)_\sharp \mathbb{P}$, where $\varphi(\cdot; \theta)$ is a differentiable invertible mapping on $\mathbb{R}^d$, depending on $\theta$.*

This assumption can be equivalently stated by the probabilistic representation $Z \overset{\mathrm{d}}{=} \varphi(U; \theta)$, where $U \sim \mathbb{P}$ (the symbol $\overset{\mathrm{d}}{=}$ denoting equality in distribution). If $\mathbb{P}$ admits a density, then so does $\mathbb{P}_\theta$ with density function given by $p_\theta(z) = |J_z\psi(z;\theta)|p(\psi(z;\theta))$ where $\psi(\cdot; \theta) = \varphi^{-1}(\cdot; \theta)$. In this last formula, $J_z\psi(z;\theta)$ refers to the Jacobian matrix where $[J_z\psi(z;\theta)]_{ij} = \frac{\partial \psi(z;\theta)_i}{\partial z_j}$.

From an abstract point of view, such a representation of a parametrized family of distributions exists under very general conditions. However, the above model is more interesting in scenarios where $\varphi(\cdot; \theta)$ is a simple operator, for instance a linear one, as is the case in most of the examples considered by Perdomo et al. [2020], Miller et al. [2021], and the dependence with respect to $\theta$ can also be made explicit. This representation is also modular in the sense that $\varphi$ could be chosen as the composition $\varphi(u; \theta) = \varphi_0(\varphi_1(u; \theta))$, where $\varphi_0^{-1}(Z)$ corresponds to a fixed (not depending on $\theta$) representation of $Z$ in a feature space and $\varphi_1(\cdot; \theta)$ models the performative effect *in the representation space*. Although we will not explicitly consider such cases in the rest of the paper, this representation of the performative effect is particularly attractive when using embedding tools based on kernels [Hofmann et al., 2008], neural nets (e.g., VAEs [Kingma and Welling, 2013]) or normalizing flows [Papamakarios et al., 2021, Kobyzev et al., 2021].

This structural assumption on the performative effect yields a new estimator for the performative gradient, which may be seen as an instance of the "reparametrization trick" used in VAEs, normalizing flows or by Kucukelbir et al. [2017]. Mohamed et al. [2020] also refer to this approach as "pathwise" gradient estimation.

**Theorem 1** (Performative Risk Gradient). *Under Assumption 1, the gradient of the performative risk is given by*

$$\nabla_\theta \mathrm{PR}(\theta) = \mathbb{E}_\theta \left[ \nabla_\theta \ell(Z; \theta) \right] + \mathbb{E}_\theta \left[ \mathrm{J}_\theta^T \varphi(\psi(Z; \theta); \theta) \nabla_z \ell(Z; \theta) \right], \qquad (2)$$

*where $\nabla_z \ell(z; \theta)$ and $\nabla_\theta \ell(z; \theta)$ denote respectively the gradient with respect to the first and the second parameter of the loss, and $\mathrm{J}_\theta^T \varphi(u; \theta)$ is the transpose of the Jacobian with respect to $\theta$.*

*Proof.* Notice that under assumption 1, we can rewrite the decoupled risk with a change of variable as $\mathrm{DPR}(\theta, \theta') = \mathbb{E}[\ell(\varphi(U; \theta); \theta')]$. This expression leads to the following.

$$\nabla_\theta \mathrm{PR}(\theta) = \nabla_\theta \mathbb{E}[\ell(\varphi(U; \theta); \theta)] = \mathbb{E} \left[ \nabla_\theta \ell(\varphi(U; \theta); \theta) + \mathrm{J}_\theta^T \varphi(U; \theta) \nabla_z \ell(\varphi(U; \theta)); \theta) \right],$$

which gives eq. (2) under a change of variable. $\qquad \square$

## 2.2 Estimating the Performative Gradient

From Theorem 1, it is clear that the gradient of performative risk in eq. (2) is composed of two terms – the first term corresponds to the classical risk minimization, while the second one – which we will refer to as the performative gradient in the following – captures the performative effect. The first term can be estimated by $\frac{1}{n} \sum_{i=1}^{n} \nabla \ell(Z_i; \theta)$ as usual. For the second term, we propose the following estimator.

**Definition 1** (Reparameterization-based Performative Gradient Estimator). *The performative gradient* $\nabla_\theta \mathrm{DPR}(\theta, \theta')|_{\theta'=\theta}$ *admits as unbiased estimator:*

$$\hat{G}_\theta^{RP} = \frac{1}{n} \sum_{i=1}^{n} \mathrm{J}_\theta^T \, \varphi(\psi(Z_i; \theta); \theta) \nabla_z \ell(Z_i; \theta). \tag{3}$$

This estimator allows performing gradient descent to minimize the performative risk and thus, if the performative objective is well behaved, to converge to the performative optimal point. $\hat{G}_\theta^{\mathrm{RP}}$ should be compared to the following estimator used by Izzo et al. [2022], which relies on the well-known score function formula –see [L'Ecuyer, 1991, Kleijnen and Rubinstein, 1996, Mohamed et al., 2020] and references therein.

$$\hat{G}_\theta^{\mathrm{SF}} = \frac{1}{n} \sum_{i=1}^{n} \ell(Z_i; \theta) \nabla_\theta \log p_\theta(Z_i).$$

While both $\hat{G}_\theta^{\mathrm{RP}}$ and $\hat{G}_\theta^{\mathrm{SF}}$ estimate the same quantity $\nabla_\theta \mathrm{DPR}(\theta, \theta')|_{\theta'=\theta}$, $\hat{G}_\theta^{\mathrm{RP}}$ has two distinct advantages over $\hat{G}_\theta^{\mathrm{SF}}$. First, computing $\hat{G}_\theta^{\mathrm{SF}}$ requires access to the analytical form of $p_\theta$, which is fairly unrealistic in a learning scenario, whereas our estimator $\hat{G}_\theta^{\mathrm{RP}}$ only requires knowledge of $\varphi$, paving the way for a *semi-parametric approach* in which the performative effect is modelled explicitly, but not the distribution of the data. For general maps $\varphi$, $\hat{G}_\theta^{\mathrm{RP}}$ still requires to use the inverse mapping $\psi$, however this is not required in situations where the Jacobian $\mathrm{J}_\theta \, \varphi(u; \theta)$ does not depend on $u$. Specifically, when $\varphi$ is a shift operator, one obtains a very simple expression for $\hat{G}_\theta^{\mathrm{RP}}$ as shown in the following example.

**Example 1** (Shift Operator). *If the performative effect can be modelled by a shift operator, i.e.,* $\varphi(U; \theta) = U + \Pi(\theta)$, *the* $\hat{G}_\theta^{RP}$ *estimator is given by:*

$$\hat{G}_\theta^{RP} = \mathrm{J}_\theta^T \, \Pi(\theta) \frac{1}{n} \sum_{i=1}^{n} \nabla_z \ell(Z_i; \theta),$$

*where* $\mathrm{J}_\theta \, \Pi(\theta)$ *is the Jacobian of the performative shift* $\Pi(\theta)$.

In addition to removing the need to know $p_\theta$, a second advantage of $\hat{G}_\theta^{\mathrm{RP}}$ is that it can lead to significant decrease of the variance of the estimates, as illustrated by the following example.

**Example 2** (Perfomative Gaussian Mean estimation). *Let* $\ell(z; \theta) = \|z - \theta\|^2/2$, *and* $Z \stackrel{d}{=} U + \Pi\theta$, *that is,* $\Pi(\theta) = \Pi\theta$ *is a linear shift operator. We will assume* $U \sim \mathcal{N}(0, \sigma^2 I_d)$, *so that* $p_\theta(z) \propto \exp[-\|z - \Pi\theta\|^2/(2\sigma^2)]$, *where* $\Pi$ *represents the performative effect. The gradient of* $\mathrm{DPR}(\theta, \theta')$ *w.r.t. the distributional parameter* $\theta$ *is given both by*

$$\nabla_\theta \mathrm{DPR}(\theta, \theta') = \mathbb{E}_\theta[\Pi^T \nabla_z \ell(Z; \theta')] = \Pi^T \mathbb{E}_\theta[Z - \theta'] = \Pi^T \mathbb{E}[U + a] \quad \textit{(reparameterization)}$$

$$= \mathbb{E}_\theta[\ell(Z; \theta') \nabla_\theta \log p_\theta(z)] = \Pi^T \frac{1}{2\sigma^2} \mathbb{E}\left[\|U + a\|^2 U\right] \quad \textit{(score function)}$$

*where* $a = \Pi\theta - \theta'$. *Hence, in this case* $\hat{G}_\theta^{RP} = \Pi^T \frac{1}{n} \sum_{i=1}^{n} (U_i + a)$, *while* $\hat{G}_\theta^{SF} = \Pi^T \frac{1}{2n\sigma^2} \sum_{i=1}^{n} \|U_i + a\|^2 U_i$. *Both of these expressions have equal expectation* $\Pi^T(\Pi\theta - \theta')$ *which corresponds to the gradient of* $\mathrm{DPR}(\theta, \theta')$ *w.r.t.* $\theta$. *However, the reparametrization estimator* $\hat{G}_\theta^{RP}$ *has covariance* $\sigma^2 \Pi^T \Pi/n$ *while the score-based estimator* $\hat{G}_\theta^{SF}$ *has covariance:*

$$\frac{1}{n} \Pi^T \left( \frac{(d^2 + 6d + 8)\sigma^2 + 2(d + 4)\|a\|^2 + \|a\|^4/\sigma^2}{4} I_d + aa^T \right) \Pi.$$

The details of this computation can be found in appendix A.1. Both estimators are unbiased but note that while $\hat{G}_\theta^{\text{RP}}$ would always be an unbiased estimator of the performative gradient without any further assumption on the distribution of $U$, the unbiasedness of $\hat{G}_\theta^{SF}$ relies on the fact that $U$ is Gaussian. $\hat{G}_\theta^{RP}$ has a covariance that does not depend on $\theta$, $\theta'$ nor on the dimension $d$. In contrast, $\hat{G}_\theta^{SF}$'s covariance includes a factor that increases with $d^2$, making it unreliable in high dimensions. It also includes additional terms that grow with the norm of $\Pi\theta - \theta'$, so the estimator becomes less reliable when the performative effect is strong.

One could argue that the previous result does not provide a fair comparison between both estimators, as $G^{SF}$'s variance can be reduced by subtracting a baseline. Indeed, $\tilde{G}_\theta^{\text{SF}} = \frac{1}{n}\sum_{i=1}^n (\ell(Z_i;\theta) - m)\nabla_\theta \log p_\theta(Z_i)$ is also an unbiased estimator of the gradient of the performative effect (for any choice of the baseline $m$), as the score function has, by definition, zero expectation. Tuning $m$ properly may reduce the variance by creating a so-called *control variate* —see, e.g., Greensmith et al. [2004] for the use of this principle in policy gradient methods. However, similar calculations (detailed in appendix A.1) show that the minimum covariance that can be achieved by subtracting a baseline is

$$\frac{1}{n}\Pi^T \left( \left( (1 + d/2)\sigma^2 + \|a\|^2 \right) I_d + aa^T \right) \Pi,$$

which is still larger than the covariance of $\hat{G}_\theta^{\text{RP}}$ by a factor that grows with the model dimension $d$. The fact that the reparameterization-based estimator is preferable when considering the Gaussian distribution with quadratic loss function was observed before by Mohamed et al. [2020] in the scalar case. The above computations however show that the difference between the two approaches gets more and more significant as the dimension increases. The case of other distribution/loss function combination still needs to be investigated.

## 3 Classification under Performative Shift

In this section, we specialize to the setting of binary classification which encompasses various machine learning applications where performative effects are expected. Usually, this setting involves a desirable class and an undesirable one. For example, the desirable class might represent college admission, loan acceptance, no-spam email, or probation. In this setting, one can also expect that individuals belonging to the favored class – we designate this as class $1$ – do not need to alter their features, or only with small changes. On the contrary, individuals with negative predictions – in class $0$ – have an incentive to modify their features, resulting in a significant performative effect.

We particularize the arguments introduced in section 2 to the setting of binary classification, by fixing $z = (x, y)$, with a covariate vector $x \in \mathbb{R}^d$ and a label $y \in \{0, 1\}$. As is done classically —see, e.g., [Bach, 2024], we further assume that the classifier $f_\theta(x)$ is a real valued function that depends on a parameter $\theta \in \mathbb{R}^p$ and that a convex loss surrogate $\Phi$ is used, such that the loss function $\ell(z; \theta)$ is equal to $\Phi((-1)^y f_\theta(x))$. We model the performative effect as label-dependent push forward models, i.e., that, under $\mathbb{P}_\theta$,

$$X|_{Y=1} \stackrel{\text{d}}{=} \varphi_1(U_1; \theta) \text{ and } X|_{Y=0} \stackrel{\text{d}}{=} \varphi_0(U_0; \theta),$$

where $\varphi_1$ and $\varphi_0$ represent the performative changes affecting class-conditional distributions of classes 0 and 1 respectively. If the classifier $f_\theta(x)$ is sufficiently expressive, changes such that $\varphi_1 = \varphi_0$ will not create performative effects. We thus focus on scenarios where the *performative changes affect each class-dependent distribution differently*. For concreteness, we will assume the following.

**Assumption 2.** $\mathbb{P}_\theta(Y = 1) = \rho$ *is fixed and not subject to performative effects.*

**Assumption 3.** $\varphi_1$ *does not depend on $\theta$, and for simplicity, we assume it is the identity function.*

**Assumption 4.** $\varphi_0(u_0; \theta) = u_0 + \Pi(\theta)$ *is a shift operator.*

Assumption 2 is a consequence of the intuitive property that even if the distribution is modified, the ground truth labels are not impacted by the performative effect. Assumption 3 allows to focus on the performative effect on the unfavored class and simplifies the presentation but could be easily relaxed. Finally, assumption 4 is restricting the performative change to a shift, which does simplify the problem but still corresponds to a realistic model for feature alteration.

It is important to stress that, despite the fact that this performative effect is modelled as a shift, the joint distribution of $Z = (X, Y)$ does not belong to the location-scale family discussed by Miller et al. [2021]. Under these assumptions, the decoupled performative risk takes the following form:

$$\text{DPR}(\theta, \theta') = \mathbb{E}_\theta \left[ \Phi \left( (-1)^Y f_{\theta'}(X) \right) \right] = \rho \mathbb{E} \left[ \Phi(f_{\theta'}(U_1)) \right] + (1-\rho) \mathbb{E} \left[ \Phi(-f_{\theta'}(U_0 + \Pi(\theta))) \right], \quad (4)$$

where the performative effect is only manifested in the second term, which corresponds to class 0.

**Remark 1** (Localization of the Performative Shift). *In eq. (4), we refer to $U_0$ which corresponds to the covariates of the second class in the absence of performative effect, i.e., when $\theta = 0$. However, for a shift operator, for any value of $\bar{\theta}$, one may equivalently write that, under $\mathbb{P}_\theta$, $X \overset{d}{=} \varphi(U_{\bar{\theta}}; \theta)$, where $\varphi(u; \theta) = u + \Pi(\theta) - \Pi(\bar{\theta})$ and $U_{\bar{\theta}}$ is distributed under $\mathbb{P}_{\bar{\theta}}$. Thus eq. (4) can equivalently be rewritten by taking expectation under an arbitrary parameter value $\bar{\theta}$, upon defining the performative effect as $U_{\bar{\theta}} + \Pi(\theta) - \Pi(\bar{\theta})$ and the linear model as $f_\theta(x) = x^T(\theta - \bar{\theta})$.*

## 4 Convexity of Performative Risk

Identification of the cases in which the performative risk $\text{PR}(\theta)$ is convex is an important step towards generalizing results obtained in the context of traditional (ie., non performative) learning theory. Existing results mainly exploit the fact that if the loss function is strongly convex, a sufficiently small performative effect cannot break this convexity. For this reason, it is often assumed [Miller et al., 2021, Hardt and Mendler-Dünner, 2023] that the change in the distributions has a bounded sensitivity with respect to the parameters, using the 1-Wasserstein distance:

$$W_1 \left( P_\theta, P_{\theta'} \right) \leqslant \varepsilon \left\| \theta - \theta' \right\|_2 .$$

Note that in our setting, such $\epsilon$ exists and corresponds to the operator norm of $\Pi(\theta)$. In order to preserve convexity, it is then needed that $\epsilon \leqslant \mu/2L$ when the risk is $L$-smooth and $\mu$-strongly convex. The pricing model, considered by Izzo et al. [2022], is a very simple example showing that the performative risk can be convex while not fulfilling this criterion.

**Example 3** (Pricing Model). *Given a fixed set of $d$ resources, the pricing model aims at finding the prices $\theta \in \mathbb{R}^d$ for the $d$ ressources that maximize the overall profit given the elasticity of the demand level:*

$$\ell(z; \theta) = -z^T \theta \text{ with } Z \overset{d}{=} \varphi(U; \theta) = U - \Pi\theta, \quad (5)$$

*where $\Pi$ is a diagonal matrix with positive elements, encoding the elasticity of the demand level to a raise in price of each resource, and $\mu = \mathbb{E}[U]$ contains the baseline demand levels for each resource.*

In this example, the performative risk $\text{PR}(\theta) = -\sum_{i=1}^d (\mu_i - \Pi_{ii}\theta_i)\theta_i$ is a strongly convex quadratic function minimized at $\theta_i^\star = \mu_i/(2\Pi_{ii})$. In contrast, the decoupled performative risk $\text{DPR}(\theta, \theta') = -(\alpha - \Pi\theta)^T \theta'$ is still convex, but not strongly convex in $\theta$ and is always minimized in $\theta'$ at infinity. Despite its simplicity, this example is thus not covered by existing theorems, and retraining procedures considered by Perdomo et al. [2020] fail by diverging.

Moreover, this example highlights that requiring a small sensitivity for the performative effect does not match the true convexity conditions. The performative risk is indeed strongly convex as long as the $\Pi_{ii}$ are positive, irrespectively of their magnitude. In contrast, in this example, the performative risk would become non-convex if one of the $\Pi_{ii}$ were negative, even with a small magnitude.

This motivates the search for related phenomenons in the classification context, by looking at conditions for ensuring the convexity of eq. (4). Indeed, we show that in the classification setting, one can also observe convexity without restriction on the magnitude of the performative effect. For this, we consider the case of linearly parameterized models, which we denote for simplicity by $f_\theta(x) = x^T \theta$. Note that, as discussed in Section 2, our model of performative effect is composable and hence, we could also consider the more general linearly parameterized model in which $f_\theta(x) = \psi(x)^T \theta$, with a linear-in-the-parameter performative effect in the feature space such that $\psi(X) = U + \Pi(\theta)$. For ease of notation, we stick to the case where $\psi$ is the identity function in the following. Using standard arguments, the choice of a convex loss surrogate $\Phi$ then entails that $\text{DPR}(\theta, \theta')$ is a convex function of $\theta'$. For the same reason, if $\Pi(\theta) = \Pi\theta$ is a linear-in-parameters shift operator, $\text{DPR}(\theta, \theta')$ is also convex in $\theta$. Note however that, unless there is no performative effect (i.e., if $\Pi = 0$), eq. (4) is not jointly convex in $(\theta, \theta')$. The following result show that it is nonetheless the case that the performative risk $\text{PR}(\theta) = \text{DPR}(\theta, \theta)$ is convex under the condition that $\Pi$ is a positive semidefinite matrix (see proof in appendix A.2).

**Theorem 2** (Convexity of Classification Performative Risk). *Under assumptions 2 to 4, for linearly parameterized classifier $f_\theta(x) = x^T\theta$ and linear shift operator $\Pi(\theta) = \Pi\theta$, the performative risk $\mathrm{PR}(\theta)$ is convex when $\Pi$ is a positive semidefinite matrix and one of the following conditions holds.*

*(a) $\Phi$ is the quadratic loss function;*

*(b) $\Phi$ is a convex non increasing function (such as hinge, logistic or exp loss).*

This theorem allows us to extend the known convexity results to losses that are not strongly convex, and to performative effects with arbitrary magnitude.

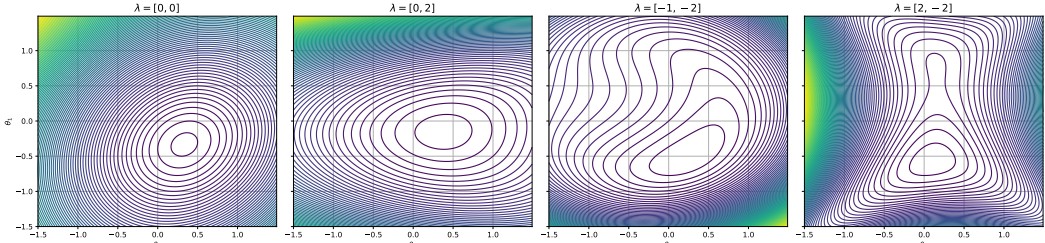

Figure 1: Profile risk for classifying two Gaussian centered in $\mu_0 = (0,0)$ and $\mu_1 = (-1,1)$ with quadratic loss and various values of $\lambda$ for the diagonal coefficients of $\Pi$. The performative risk remains convex as long as $\Pi$ is positive semidefinite i.e. $\lambda \geq 0$, and becomes non-convex whenever some of the $\lambda_i$ are negative.

**Remark 2** (Generalization to Performative Effect Affecting Both Classes). *One could remove assumption 3 to allow class 1 to change under performative effect. The convexity of $\mathrm{PR}(\theta)$ remains if $\varphi_1(u;\theta) = u - \Pi_1\theta$, where $\Pi_1$ is a positive semidefinite matrix. Similarly, the classification task becomes harder with performative effects and the performative risk is convex.*

## 5  Connection with Robustness and Regularization

In order to enforce strong convexity of the loss function $\ell(\cdot;\theta)$, previous works on performative prediction have considered the use of an additional regularization term —see, e.g., Section 5.2 of [Perdomo et al., 2020] where logistic regression is used with a ridge regularizer. When doing so, it has been observed empirically that the retraining method performs quite well. To build on this observation, we show below that for linear-in-the-parameter performative effects that tends to make the classification task harder, the performative optimum may indeed be interpreted as a regularized version of the base classification problem. This regularization does not take the form of and additive penalty but can be interpreted as the solution of a specific adversarially robust classification objective. In this section, we use the slightly stronger assumption that $\Pi$ is a symmetric positive definite matrix, in order to ensure that both $\|v\|_\Pi = (v^T\Pi v)^{1/2}$ and $\|v\|_{\Pi^{-1}} = (v^T\Pi^{-1}v)^{1/2}$ are norms on $\mathbb{R}^d$.

**Theorem 3** (Variational Formulation of the performative Risk). *Under assumptions 2 to 4, for linearly parameterized classifiers $f_\theta(x) = x^T\theta$ and linear shift operators $\Pi(\theta) = \Pi\theta$, and assuming that $\Phi$ is a convex non increasing function and that $\Pi$ is symmetric positive definite, the performative risk may be rewritten as*

$$\mathrm{PR}(\theta) = \rho\mathbb{E}[\Phi(U_1^T\theta)] + (1-\rho)\mathbb{E}\left[\max_{\{\Delta U_0 : \|\Delta U_0\|_{\Pi^{-1}} \leq \|\theta\|_\Pi\}} \Phi(-(U_0 + \Delta U_0)^T\theta)\right]. \quad (6)$$

Intuitively, for a classification-calibrated loss function [Bartlett et al., 2006, Bach, 2024] and classes with identical covariances, we expect $\theta$ to align with the direction of $\mu_1(\theta) - \mu_0(\theta)$, so that, when $\Pi$ is positive definite, the performative shift $\Pi\theta$ has itself a positive dot product with $\mu_1(\theta) - \mu_0(\theta)$. The reformulation of the performative risk in eq. (6) formalizes this intuition by showing that the performative optimum is associated to an adversarially robust classification task [Goodfellow et al., 2015, Madry et al., 2018, Ribeiro et al., 2023] in which the points of class 0 are allowed to shift towards those of class 1, so as to increase the overall loss. Compared to objectives found in the robust

classification literature, the specificity of eq. (6) lies in the fact that the tolerance (or budget) on the adversarial displacement $\Delta U_0$ depends on both $\Pi$ and $\theta$.

To understand the role played by the $\|\cdot\|_\Pi$ and $\|\cdot\|_{\Pi^{-1}}$ norms, consider the particular case where only a subset of the variables have a performative effect, i.e., if we let $\theta$ and $U_0$ be partitioned into

$$\theta = \begin{pmatrix} \theta_p \\ \cdots \\ \theta_s \end{pmatrix} \text{ and } U_0 = \begin{pmatrix} U_{0,p} \\ \cdots \\ U_{0,s} \end{pmatrix},$$

with $\Pi_p = \gamma I$ and $\Pi_s = \epsilon I$, one obtains, letting $\epsilon$ tend to zero, that the performative risk is equal to

$$\mathrm{PR}(\theta) = \rho \mathbb{E}[\Phi(U_1^T \theta)] + (1-\rho)\mathbb{E}\left[ \max_{\{\Delta U_{0,p} : \|\Delta U_{0,p}\| \leq \gamma \|\theta_p\|\}} \Phi\left(-(U_{0,s}^T \theta_s + (U_{0,p} + \Delta U_{0,p})^T \theta_p)\right)\right].$$

The above expression shows that in this case, only the coordinates subject to the performative effect appear in the adversarial reformulation.

In the proof of Theorem 3 (see appendix A.3), we observe that the second term of eq. (6) may also be rewritten as $(1-\rho)\mathbb{E}[\Phi(-U_0^T \theta - \|\theta\|_\Pi^2)]$. Similarly to the case studied by Ribeiro et al. [2023], the term $\|\theta\|_\Pi^2$ that appears inside the surrogate loss function $\Phi$ has a regularization effect. Note however that it is not equivalent to the use of a standard ridge regression penalty on $\theta$. The following theorem provides a bound on the performative optimum that highlights the role played by $\Pi$ on the significance of this regularization effect.

**Theorem 4** (Regularization Bound). *Define $\mu_i = \mathbb{E}[U_i]$. Under assumptions 2 to 4, for linearly parameterized classifiers $f_\theta(x) = x^T\theta$ and linear drift operators $\Pi(\theta) = \Pi\theta$, when $\Phi$ is a convex non increasing function and $\Pi$ a symmetric positive definite matrix, the minimizer $\theta^*$ of $\mathrm{PR}(\theta)$ satisfies the following condition.*

$$\|\theta^*\|_\Pi \leq \frac{\|\Pi^{-\frac{1}{2}}(\rho\mu_1 - (1-\rho)\mu_0)\|}{1-\rho}. \tag{7}$$

Theorem 4 shows that the performative optimum has a smaller value in $\|\cdot\|_\Pi$ norm when the performative effect is stronger, that is, when $\Pi$ gets larger. In the particular case where $\Pi = \gamma I$, eq. (7) rewrites as $\gamma^{1/2}\|\theta^*\| \leq \gamma^{-1/2}\|\rho\mu_1 - (1-\rho)\mu_0\|/(1-\rho)$ and thus larger values of $\gamma$ decrease the r.h.s. while the l.h.s. increases for identical values of $\|\theta^*\|$, showing that $\|\theta^*\|$ has to decrease to zero.

## 6 Experiments

In this section[1], we test the performance of our algorithm Reparametrization-based Performative Gradient (RPPerfGD) with respect to existing algorithms. Three baselines were introduced in Perdomo et al. [2020]. First, *Repeated Risk Minimization (RRM)* computes at each step the next $\theta$ to minimize the non-performative risk, leading to the update rule $\theta^{t+1} = \arg\min_{\theta'} \mathrm{DPR}(\theta^t, \theta')$. In practice, we found that this algorithm is unstable as soon as performative effects become significant. We thus report separately the results obtained with this algorithm in appendix B.2. A second baseline is *Repeated Gradient Descent (RGD)*, which ignores the performative effect but limits itself to a gradient step towards this minimization $\theta^{t+1} = \theta^t - \eta \nabla_{\theta'} \mathrm{DPR}(\theta, \theta')|_{\theta'=\theta}$. In numerical experiments, it is often chosen to add a regularizer to the objective function, which is particularly interesting in the context of performative learning, as discussed in section 5. Hence, we report *Regularized Repeated Gradient Descent (RRGD)*, that corresponds to the repeated gradient with a loss function including an additional ridge penalty on $\theta$, leading to a more conservative behavior that is also more robust to performative effects. Finally, we also compare to the *Score Function Performative Gradient Descent (SFPerfGD)* which estimates the performative part of the gradient using the $\hat{G}_\theta^{\mathrm{RP}}$ estimator based on the score-function approach (see section 2). This was previously used in small dimensions in Miller et al. [2021].

---

[1] The code is available at `https://github.com/totilas/PerfOpti`

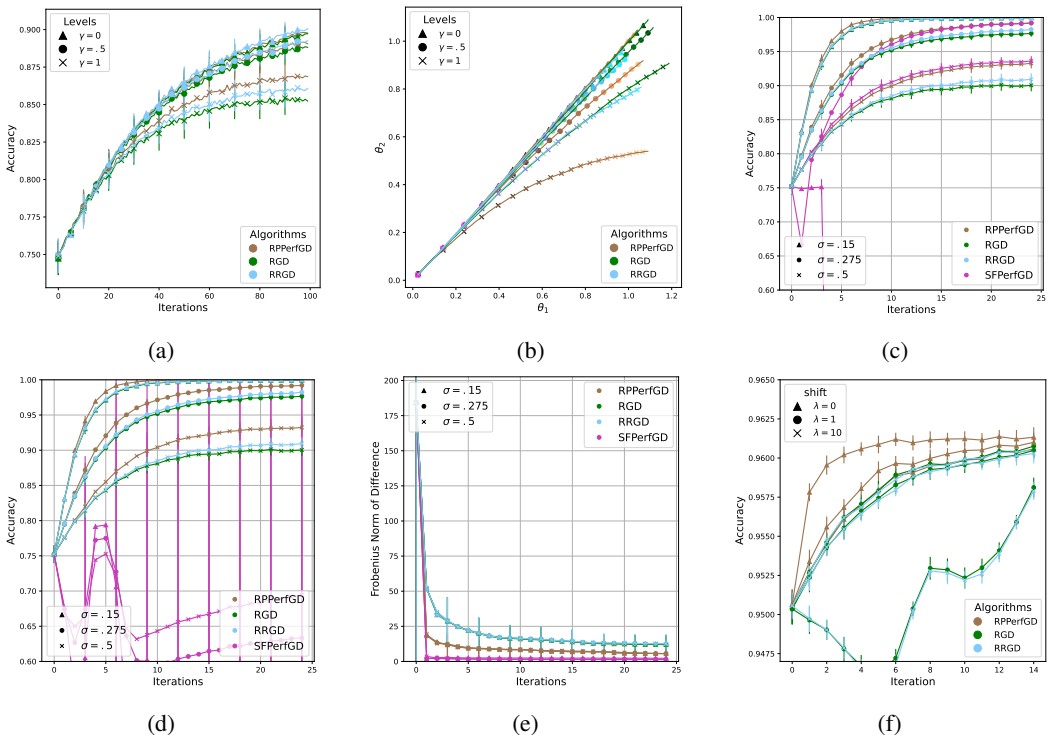

Figure 2: **(a)** Logistic regression to classify two Gaussian distributions centered in $(0, 0)$ and $(-1, -1)$ and different magnitudes of performative effects $\gamma$. We report the accuracy for three different magnitudes of the performative effects, from no performative effect ($\gamma = 0$) to a strong one ($\gamma = 1$). **(b)** we report the position of the parameter $\theta$ in its 2D-space, starting from $(0, 0)$ and following different paths depending on the algorithm. **(c)** Accuracy of a classification with quadratic loss on two Gaussian distributions of dimension 7 with various levels of variance $\sigma$ of the distributions. **(d)** Same experiments but using the learnt $\Pi$ for RPPerfGD. **(e)** In this case, distance between the true matrix $\Pi$ and the estimated version. Note that in RGD and RRGD the estimation of $\Pi$ is not used in the algorithm. **(f)** Logistic regression for the Housing dataset with various magnitude of performative shift $\lambda$ on the coordinates 0, 4 and 6. Accuracy is averaged over 20 runs.

**Influence of the performative effect** In fig. 2b, we illustrate how taking into account the performative effect allows to mitigate regimes with strong distribution shift. We generate two Gaussian distributions, one fixed for class 1 and one moving with mean $\mu(\theta)^T = -(1, 1) + \gamma \theta^T \operatorname{diag}(0.1, 0.9)$, where $\gamma$ is the magnitude parameter effect. We learn a logistic regression and report the accuracy of the predictions as well as the trajectory of the parameter in $(\theta_1, \theta_2)$-space. As expected, when there is no performative effect ($\gamma = 0$), all methods are equivalent. As soon as there are performative effects, Performative gradient takes advantage over other methods. RRGD proposes an interesting tradeoff in terms of performance: agnostic to the performative effect, it still moderates its value and thus the magnitude of the performative effect.

**Stability of the estimator** In fig. 2c, we illustrate the result of example 2, by training a classifier with the square loss, showing that the score-based estimator used in SFPerfGD becomes unstable in high dimensions. We use Gaussian distributions of dimension 7 with two dimensions subject to performative effects. We vary the variance of the distributions. When the scale $\sigma$ is small, the variance of the estimator increases to the point of making learning impossible with unstable trajectories of the parameter $\theta$. Even when the scale is small enough to ensure convergence, RPPerfGD provides faster convergence illustrating its better scalability for high dimensions.

**Estimation of $\Pi$** We estimate $\Pi$ in fig. 2d and fig. 2e by running a ridge regression along the successive deployments of the model as described in Algorithm 1: the ridge penalty ensures that initially the estimate of $\Pi$ is close to zero making the RPPerfGD updates very similar to those of

RGD and it is easy to check that order $d$ deployments are enough to obtain a non void estimate of $\Pi$. While this plug-in approach is not guaranteed to converge from a theoretical standpoint, we observe results that very similar to the case where $\Pi$ is fully known.

**Houses price prediction**    To simulative performative effects from a dataset, we follow the methodology of Perdomo et al. [2020], by shifting the coordinate $i$ of a factor $\lambda\theta_i$ if the $i$-th coordinate could be easily modified, and keeping its real value intact otherwise. We use the binarized version of the Housing dataset[2], where the outcome is whether the price is high or not. Assuming that a seller wants to obtain a high price, the high price is the favored class. Some characteristics are harder to tamper with such as the location or the income, whereas other can be slightly adjusted such as the household and the number of bedrooms (a room could be promoted bedroom). Coordinates 0, 4 and 6 are thus shifted while other remains identical. We see that when the magnitude of the shift increases, RPPerfGD outperforms RGD. In particular, it seems that RPPerfGD succeeds in converging faster than the non performative approach.

---

**Algorithm 1:** RPPerfGD with $\Pi$ learning

---

**Input**    : Stepsize $\eta$, regularizer $\lambda$, starting $\theta_0$, Loss $\ell$
**Output** : Parameters $\theta_K$ and diagonal matrix $\Pi_K$

1  $\Pi_0 \leftarrow 0_{d \times d}$ `// initialize` $\Pi$ `as a zero matrix of size` $d \times d$
2  **for** $k \in \{0, \ldots, K-1\}$ **do**
3  $\quad$ Receive $n$ samples $\{x_k^i\}_{i=1}^n \sim D(\theta_k)$ with $n_0$ samples of label $-1$ denoted $(x_{0,k})_k$
4  $\quad$ Compute $\nabla_1 \leftarrow \frac{1}{n}\sum_{i=1}^n \nabla_\theta \ell(x_k^i, \theta_k)$`// Non performative part of the gradient`
5  $\quad$ Compute $\nabla_2 \leftarrow \frac{1}{n}\Pi_k^\top \sum_{i=1}^{n_0} \nabla_x \ell(x_{0,k}^i, \theta_k)$`// Performative part of the gradient`
$\quad\quad$ `over negative samples`
6  $\quad$ $\theta_{k+1} \leftarrow \theta_k - \eta(\nabla_1 + \nabla_2)$ `// Gradient Descent step`
7  $\quad$ $\Pi_{k+1} \leftarrow \arg\min \sum_{j=1}^k \sum_{l=1}^{n_0} \|x_{0,j}^l - \hat{\mu} - \Pi\theta_j\|^2 + \lambda\|\Pi\|^2$ `//` $\hat{\mu}$ `is the estimated`
$\quad\quad$ `mean of the class`

---

## 7   Conclusion

In this work, we have investigated the consequences of assuming a novel, more explicit, model for performative effects under the form of a push-forward shift of distribution. We have demonstrated that it comes with practically important consequences, such as enabling more reliable performative gradient estimation in large dimensional models.

In the classification case, we observed that when the change of distribution is given by a linear-in-parameters shift, the performative risk is convex under relatively general assumptions. It would be interesting to study how these results may extend to non-linear models for the performative effect.

Finally, we have shown that certain kinds of performative effect induce implicit regularization of the risk minimization problem. Moreover, this regularization effect can alternatively be viewed through the lens of adversarial robustness. It would be useful to explore whether this reformulation can be used to optimize the performative risk without an explicit model for the performative effect.

## 8   Acknowledgments

This work was supported by grants ANR-20-THIA-0014 program "AI PhD@Lille" and ANR-22-PESN-0014 under the France 2030 program. The authors thank Francis Bach, Bruno Loureiro and Kamélia Daudel for their helpful insights.

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

# A  Supplementary Material

## A.1  Proofs for Example 2

Using the notations of Example 2 the score function estimator may we written as $\Pi^T \frac{1}{2\sigma^2} G$, where $G = \|U + a\|^2 U$ with $U \sim \mathcal{N}(0, \sigma^2 I_d)$ and $a \in \mathbb{R}^d$ is a deterministic vector depending on the parameters $\theta, \theta'$ and the performative effect. To compute the expectation and covariance matrix of $G$ we will use Isserlis' (or Wick's probability) theorem which state that

(a) $\mathbb{E}[U_{i_1} \ldots U_{i_{2m+1}}] = 0$, for any $\{i_1, \ldots, i_{2m+1}\} \in \{1, \ldots, d\}^{2m+1}$

(b)
$$\mathbb{E}[U_{i_1} \ldots U_{i_{2m}}] = \sigma^{2m} \sum_{\{j_1, k_1\}, \ldots, \{j_m, k_m\} \in \mathcal{P}(\{i_1, \ldots, i_{2m}\})} \delta_{j_1 k_1} \ldots \delta_{j_m k_m}$$

where $\mathcal{P}(\{i_1, \ldots, i_{2m}\})$ denotes all the distinct ways of partitioning $\{i_1, \ldots, i_{2m}\}$ into non-overlapping (unordered) pairs and $\delta$ is the Kronecker delta. It is easily checked that the number of partitions in $\mathcal{P}(\{i_1, \ldots, i_{2m}\})$ is equal to $\binom{2m}{m} \frac{m!}{2^m}$ which is also equal to the product of all odd numbers between 1 and $2m - 1$.

For the expectation, $\mathbb{E}[G] = \mathbb{E}[(\|U\|^2 + \|a\|^2 + 2a^T U)U] = 2\mathbb{E}(UU^T)a = 2\sigma^2 a$, as the expansion of all other terms would involve and odd number of coordinates of $U$.

Let $M = \mathbb{E}[GG^T]$, we have

$$M_{ij} = \mathbb{E}\left[ \sum_{k=1}^d (U_k + a_k)^2 \sum_{l=1}^d (U_l + a_l)^2 U_i U_j \right]$$

$$= \mathbb{E}\left[ \sum_{k=1}^d \sum_{l=1}^d \left( U_k^2 U_l^2 + U_k^2 a_l^2 + U_l^2 a_k^2 + 4 a_l a_k U_k U_l + a_k^2 a_l^2 \right) U_i U_j \right]$$

omitting terms in the expansion that involve and odd number of coordinates of $U$ (which have 0 expectation). Now, we apply Isserlis' theorem to each term in this decomposition, starting with the lowest order (rightmost) ones:

$$\mathbb{E}\left[ \sum_{k=1}^d \sum_{l=1}^d a_k^2 a_l^2 U_i U_j \right] = \|a\|^4 \sigma^2 \delta_{ij}$$

$$\mathbb{E}\left[ \sum_{k=1}^d \sum_{l=1}^d 4 a_l a_k U_k U_l U_i U_j \right] = 4\sigma^4 \sum_{k=1}^d \sum_{l=1}^d a_l a_k (\delta_{ij}\delta_{kl} + \delta_{ik}\delta_{jl} + \delta_{il}\delta_{jk}) = 4\sigma^4(\|a\|^2 + 2a_i a_j)\delta_{ij}$$

$$\mathbb{E}\left[ \sum_{k=1}^d \sum_{l=1}^d U_k^2 U_i U_j a_l^2 \right] = \sigma^4 \|a\|^2 \left( \sum_{k=1}^d (\delta_{ij} + 2\delta_{ki}\delta_{kj}) \right) = \sigma^4 \|a\|^2 (d+2)\delta_{ij}$$

$$\mathbb{E}\left[ \sum_{k=1}^d \sum_{l=1}^d U_k^2 U_l^2 U_i U_j \right] = \sigma^6 \left( \delta_{ij} + 2\delta_{il}\delta_{jl} + 2\delta_{ik}\delta_{jk} + 2\delta_{kl}\delta_{ij} + 8\delta_{kl}\delta_{ki}\delta_{ij} \right) = \sigma^6 (d^2 + 6d + 8)\delta_{ij}$$

where the last decomposition is obtained by examination of the $\binom{6}{3}\frac{3!}{2^3} = 15$ possible partitions of $\{k, k, l, l, i, j\}$ in 3 pairs of indices. Putting all together, one obtains

$$M = \left( (d^2 + 6d + 8)\sigma^6 + 2(d+4)\|a\|^2 \sigma^4 + \|a\|^4 \sigma^2 \right) I_d + 8\sigma^4 aa^T$$

which yields

$$\mathrm{Cov}(G) = \left( (d^2 + 6d + 8)\sigma^6 + 2(d+4)\|a\|^2 \sigma^4 + \|a\|^4 \sigma^2 \right) I_d + 4\sigma^4 aa^T \tag{8}$$

Subtracting a scalar baseline $m$ yields the estimator $\tilde{G} = (\|U + a\|^2 - m)U$ which has the same expectation as $G$. In terms of covariances, one has

$$\mathrm{Cov}(\tilde{G}) = \mathrm{Cov}(G) + m^2 I_d - 2m\mathbb{E}\left( \|U + a\|^2 UU^T \right)$$

The rightmost expression, when expanded, features terms have already been met in the computation above and it is easy to check that

$$\mathbb{E}\left(\|U + a\|^2 U U^T\right) = ((d + 2)\sigma^4 + \|a\|^2\sigma^2)I_d$$

Hence,

$$\text{Cov}(\tilde{G}) = \text{Cov}(G) + \left(m^2 - 2m((d + 2)\sigma^4 + \|a\|^2\sigma^2)\right)I_d$$

In the above equation, the scalar term $m^2 - 2m((d + 2)\sigma^4 + \|a\|^2\sigma^2)$ is minimized by choosing $m = (d + 2)\sigma^2 + \|a\|^2$ and is equal to $-\left((d + 2)\sigma^2 + \|a\|^2\right)^2\sigma^2$, which, combined with eq. (8) yields

$$\text{Cov}(\tilde{G}) \geq \left(2(d + 2)\sigma^6 + 4\|a\|^2\sigma^4\right)I_d + 4\sigma^4 a a^T \tag{9}$$

## A.2 Proof of Theorem 2

*Proof.* For (a), eq. (4) may be rewritten as

$$\text{PR}(\theta) = \rho\mathbb{E}[(U_1^T\theta - 1)^2] + (1 - \rho)\mathbb{E}[((U_0 + \Pi\theta)^T\theta + 1)^2]$$

Denoting $\mathbb{E}(U_i) = \mu_i$ and $\text{Cov}_\theta(U_i) = \Sigma_i$, for $i \in \{0, 1\}$, one has

$$\text{PR}(\theta) = \rho\mathbb{E}[((U_1 - \mu_1)^T\theta - (1 - \mu_1^T\theta))^2] + (1 - \rho)\mathbb{E}[((U_0 - \mu_0)^T\theta + (\mu_0 + \Pi\theta)^T\theta + 1)^2]$$
$$= \rho[\|\theta\|_{\Sigma_1}^2 + (1 - \mu_1^T\theta)^2] + (1 - \rho)[\|\theta\|_{\Sigma_0}^2 + ((\mu_0 + \Pi\theta)^T\theta + 1)^2]$$

Both squared norms are convex as well as the squares of, respectively, an affine function and a convex second order polynomial (as $\Pi$ is positive semidefinite).

For (b), examining

$$\text{PR}(\theta) = \rho\mathbb{E}[\Phi(U_1^T\theta)] + (1 - \rho)\mathbb{E}[\Phi(-(U_0 + \Pi\theta)^T\theta)] \tag{10}$$

one observes that

- $\Phi(u_1^T\theta)$ is convex by our assumption on $\Phi$ (for any value of $u_1$);
- $(u_0 + \Pi\theta)^T\theta$ is a convex second order (multivariate) polynomial in $\theta$ when $\Pi$ is positive semidefinite and $v \mapsto \Phi(-v)$ is convex non decreasing, hence $\Phi(-(u_0 + \Pi\theta)^T\theta)$ is also convex.

Thus $\text{PR}(\theta)$ is also convex in $\theta$ as the expectation of convex functions. □

## A.3 Proof of Theorem 3

*Proof.* To obtain eq. (6), as $v \mapsto \Phi(-v)$ is non decreasing, one has

$$\max_{\{\Delta u_0 : \|\Delta u_0\|_{\Pi^{-1}} \leq \|\theta\|_\Pi\}} \Phi\left(-(u_0 + \Delta u_0)^T\theta\right) = \Phi\left(-u_0^T\theta - \max_{\{\Delta u_0 : \|\Delta u_0\|_{\Pi^{-1}} \leq \|\theta\|_\Pi\}} (\Delta u_0)^T\theta\right)$$

for any outcome $u_0$ of the random variable $U_0$. The maximization occurs for $\Delta u_0 = \Pi\theta$, which does not depend on $u_0$, leading to $(\Delta u_0)^T\theta = \|\theta\|_\Pi^2$ and thus

$$\max_{\{\Delta U_0 : \|\Delta U_0\|_{\Pi^{-1}} \leq \|\theta\|_\Pi\}} \Phi\left(-(U_0 + \Delta U_0)^T\theta\right) = \Phi\left(-U_0^T\theta - \|\theta\|_\Pi^2\right)$$

whose expectation is recognized as the second term of eq. (10). □

## A.4 Proof of Theorem 4

*Proof.* Recall from the proof of Theorem 2 that the performative risk can be rewritten as follows.

$$\text{PR}(\theta) = \rho\mathbb{E}[\Phi(U_1^T\theta)] + (1 - \rho)\mathbb{E}[\Phi(-(U_0 + \Pi\theta)^T\theta)]$$
$$= \rho\mathbb{E}[\Phi(U_1^T\theta)] + (1 - \rho)\mathbb{E}[\Phi(-U_0^T\theta - \|\theta\|_\Pi^2)]$$

Let $\mu_\rho = \rho\mu_1 - (1-\rho)\mu_o$. We have,

$$\Phi(0) = \mathrm{PR}(0) \geq \mathrm{PR}(\theta^*)$$
$$= \rho\mathbb{E}[\Phi(U_1^T\theta^*)] + (1-\rho)\mathbb{E}[\Phi(-U_0^T\theta^* - \|\theta^*\|_\Pi^2)]$$
$$\geq \rho\Phi(\mathbb{E}[U_1^T\theta^*]) + (1-\rho)\Phi(\mathbb{E}[-U_0^T\theta^* - \|\theta^*\|_\Pi^2])$$
$$= \rho\Phi(\mu_1^T\theta^*) + (1-\rho)\Phi(-\mu_0^T\theta^* - \|\theta^*\|_\Pi^2)$$
$$\geq \Phi\left(\rho\mu_1^T\theta^* - (1-\rho)(\mu_0^T\theta^* + \|\theta^*\|_\Pi^2)\right)$$
$$= \Phi\left(\mu_\rho^T\theta^* - (1-\rho)\|\theta^*\|_\Pi^2\right)$$

where we have successively used Jensen's inequality and the convexity of $\Phi$. Since $\Phi$ is non-increasing, we must have $0 \leq \mu_\rho^T\theta^* - (1-\rho)\|\theta^*\|_\Pi^2$. Denoting $\beta = \Pi^{\frac{1}{2}}\theta^*$, it holds that

$$(1-\rho)\|\beta\|^2 \leq \mu_\rho^T\Pi^{-\frac{1}{2}}\beta \leq \|\Pi^{-\frac{1}{2}}\mu_\rho\|\|\beta\|$$

where that last inequality is obtained using Cauchy–Schwarz. Hence, $\|\theta^*\|_\Pi = \|\beta\| \leq \frac{\|\mu_\rho^T\Pi^{-\frac{1}{2}}\|}{1-\rho}$. $\quad\square$

# B   Numerical Experiments

## B.1   Full parameters list

In this section, we report all the parameters needed to reproduce the figures in the paper. Note that we always use the same step size for all the methods. This choice stems from the fact that the methods are equivalent (up to the regularization parameter) when there is no performative effect.

Table 1: Parameters used for figure fig. 2b

| Parameter | Value |
|---|---|
| Number of iterations (num_iter) | 100 |
| Sample size ($n$) | 1000 |
| Scale ($\sigma$) | 0.5 |
| Average number of iterations (num_iter_average) | 100 |
| Step size (step_size) | 0.1 |
| Regularization parameter ($\lambda$) | $3 \times 10^{-2}$ |

Table 2: Parameters used for figure fig. 2c

| Parameter | Value |
|---|---|
| Number of iterations (num_iter) | 25 |
| Sample size ($n$) | 1000 |
| Initial scale (scale$_0$) | 0.5 |
| Transition probability matrix ($\Pi$) | diag($[0.1, 3, 0, 0, 0, 0, 0]$) |
| Mean of class 0 ($\mu$) | $\begin{bmatrix} 1 & 2 & 0.5 & 0.5 & 0 & 0 & 0 \end{bmatrix}$ |
| Average number of iterations (num_iter_average) | 100 |
| Step size (step_size) | 0.1 |
| Regularization parameter ($\lambda$) | $10^{-1}$ |

## B.2   Repeated Risk Minimization

In this section, we report the same learning tasks as those reported in the main text, but for Repeated Risk Minimization (RRM). In every setting, as soon as the performative effect is not negligible, the technique diverges. To ensure the readability of the figure and avoid shrinking the differences between the other algorithms, we report it separately.

Table 3: Parameters used for fig. 2f

| Parameter | Value |
| --- | --- |
| Number of iterations (num_iter) | 15 |
| Sample size ($n$) | 18000 |
| Number of runs (n_runs) | 20 |
| Step size (step_size) | 0.2 |
| Regularization parameter ($\lambda$) | $5 \times 10^{-3}$ |

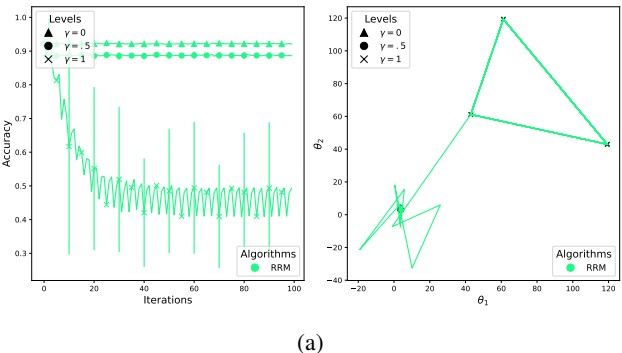

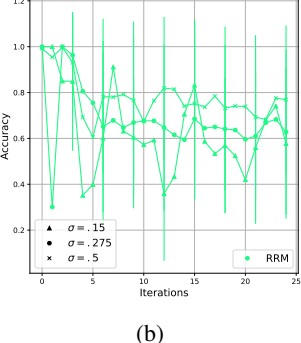

(a)

(b)

Figure 3: (a) Learning a logistic regression between two Gaussian distributions centered in $(0,0)$ and $(-1,1)$ and different magnitude of performative effects $\gamma$. (b) Accuracy of a classification with quadratic loss on two Gaussian of dimension 7 with various level of noise $\sigma$

