# OpenReview forum: "Optimal Classification under Performative Distribution Shift"
_NeurIPS.cc/2024/Conference — NeurIPS 2024 poster_

### Official Review · Reviewer_eHFF · 2024-07-11

**Soundness:** 3
**Presentation:** 2
**Contribution:** 3
**Rating:** 6
**Confidence:** 2

**Summary:**

This paper studies the performative learning problem, where the goal is to minimize some measure of *performative risk*, $PR(\theta) := \underset{\theta}{\mathbb{E}}[\ell(Z; \theta)]$, where the difficulty is that random variables come from some distribution the depends on the deployed model parameters $\theta$, i.e. $Z \sim \underset{\theta}{\mathbb{P}}$. The paper models the performative effect of some model parameters $\theta$ as a pushforward measure under some differentiable, invertible mapping that depends on $\theta$. This pushforward measure view of performative learning admits a couple of main results:

1. This gives a new expression for the performative gradient (the quantity $\nabla_{\theta} PR(\theta$)).
2. For strategic classification, a specific performative learning scenario, we find that performative risk is convex under linearity assumptions on the performative shift and the classifier model.
3. Under the same assumptions as (2), we can rewrite the performative risk as a min-max problem, connecting the performative risk in this classification scenario to adversarially robust classification.

**Strengths:**

Overall, the paper is well-written, with a couple of clarity suggestions (in "Weaknesses") that may make the presentation smoother. The technical results are sound and, to my limited knowledge of the performative learning literature, the proposal to model performative shifts as a pushforward measure seems novel and interesting. However, I must emphasize that I am not very familiar with the literature on performative learning, so I cannot judge well the impact of such an approach on existing work.

**Originality:** The main original contribution of this work is modeling the performative effect as a pushforward measure, which seems novel to my limited knowledge. The main results of the paper drop out of this modeling assumption, which seems flexible and general. However, I have a couple of questions towards how natural the specific instantiations of this pushforward measure are, particularly the "shift operator" used in many of the results (see "Questions").

**Quality:** I cannot give too informed a judgment of the quality of the results in comparison to other literature on performative learning, but to my reading, the technical results and evaluation seem sound. The authors provide a comparison to the gradient estimator for the performative effect of Izzo et al. (2022) and demonstrate their method's efficacy in comparison. The analysis seems sound, but I have no reference for how important this comparison is or whether there are better baselines in the literature to compare to.

**Clarity:** The paper is well-written overall, but I have a couple of suggestions for presentation in "Weaknesses."

**Significance:** As an outsider to the subfield, I cannot give a fully informed judgment on the significance of this paper, but, taking the comparison to Izzo et al. (2022) into account and viewing the displayed experiments, it seems that this modeling assumption does lead to an effective gradient estimator for performative risk.

**Weaknesses:**

As I am an outsider to the subfield, I cannot comment too much on the relative weaknesses of this approach to others in performative learning. I am also not aware of current results in this subfield, but it seems like optimizing this notion of performative risk is still in nascent stages if theorems like Theorem 2 exist just to show situations in which we can prove that it is convex and apply standard optimization techniques to the problem. As such, I can only give a couple of suggestions that may improve the clarity of the paper's presentation:

1. Throughout the paper, the term "performative effect" is used quite heavily, but I believe it lacks a formal definition. I assume that we are to take the performative effect as, ultimately, the distribution $\mathbb{P}_{\theta}$, but it wasn't completely clear to me on a first reading of the introduction. Explicitly defining this term in the intro may help.
2. On Page 3, when introducing the pushforward measure, I would define the "pound" symbol. I wasn't aware of this notation until I looked it up on Wikpedia.
3. Small nitpick: on page 3, "$\mathbb{P}$ admits a density" should specify the density $p(\cdot)$ that $\mathbb{P}$ admits.
4. In the Experiments section, I didn't find your algorithm "Reparametrization-based Performative Gradient (RPPerfGD)" clearly defined. I assume that the algorithm just uses the gradient in Equation 3 (Definition 1) as an estimator of the gradient and performs gradient descent, but it would be helpful to explicitly write that in the Experiments section next to the baselines.

**Questions:**

I have a couple of questions that may stem from my unfamiliarity with the literature:

1. How restrictive is the assumption that the performative effect can be modeled by a "shift operator"? I didn't fully understand how natural this assumption is, and some motivation to this assumption would help the presentation. However, I understand that this might just stem from my lack of exposure to the literature.
2. In order to estimate the gradient in Definition 1, you must have access to the form of the operator that defines the pushforward performative model, $\psi$. How realistic is it to assume that one has access to this in a non-synthetic scenario?

**Limitations:**

The authors have addressed limitations in the NeurIPS paper checklist. They also motivated the problem of performative learning in their Introduction.

---

> ### Author Rebuttal · Authors · 2024-08-06
>
> We thank the reviewer for his/her positive review and address his/her questions below. We are also grateful for the four suggestions made in the weakness section to improve the clarity of the manuscript and will made the required changes in the final version of the manuscript.
>
> **Modelling the Performative Effect as a Shift** We consider that it is indeed a key specific contribution of this paper, which was not present in earlier works on performative prediction. As noted in the paper, this assumption is however inspired by the field of adversarially robust classification where the adversary-induced distribution change is also modelled by shifts of the individual data points (where an argument also used in the proof of Theorem 3 shows that the worst case shift is indeed identical for all data points when using a linear model). In addition to exhibiting novel cases in which the performative risk can be convex in classification tasks, we believe that this assumption is also important in providing a way to interpret the effect of the performative changes. To the best of our knowledge, Theorem 2 and 3 are the first results in the literature that show that, at least in the case of linear-in-parameters shifts, if the performative change tends to bring the two class distributions closer together, then it essentially leads to a regularization effect on the parameter (when compared to a model without performative changes of distribution). We believe that this idea could be used more generically to design new approaches to performative learning.
>
> **Knowing $\varphi$ vs Knowing $p_\theta$** While we do agree that knowing $\varphi$ is a restrictive assumption, we still argue (and this is one of the main messages of the paper) that it is very different from knowing $p_\theta$. Knowing $p_\theta$ means that there is no learning problem anymore: whatever the loss function, the performative optimum could be found without training data, at worse using Monte Carlo simulations. On the other hand, even if $\varphi$ is known one still needs to use training data to find the performative optimum (as $\varphi$ does not fully define the data distributions, but only how they shift). Note also that, as argued in the paper, the shift could be taking place in an embedding space which adds even more generality. Finally, $\varphi$ could also be known up to some parameters only. In the Example of Section 6, for instance one can indeed estimate the diagonal elements of $\Pi$ using ridge regression from a very limited number of deployments and we provide in the global response PDF two additional figures showing that we need few iterations to have a good estimate of $\Pi$ and that the gradient trajectories obtained by plugging in the value of $\Pi$ estimated this way are very similar to those of RPPerfGD when using the exact value of $\Pi$.

---

> > ### Comment · Reviewer_eHFF · 2024-08-09
> >
> > I thank the authors for their comprehensive responses and willingness to address the weaknesses I pointed out in the full work. Due to my lack of exposure in the area of performative learning, I didn't initially appreciate the novelty in the model, and the response made me understand better the novelty and place in the literature. I'd like to raise my score from 5 to 6 (Weak Accept).

---

> > > ### Author Response · Authors · 2024-08-12
> > >
> > > We thank the reviewer for taking the time to carefully assess our rebuttal and for raising his/her score.

---

### Official Review · Reviewer_uddR · 2024-07-12

**Soundness:** 2
**Presentation:** 3
**Contribution:** 2
**Rating:** 5
**Confidence:** 4

**Summary:**

This paper considers a specific performative effect that's characterized as a transformation on the original probability measure on covariate X, which is novel in the literature. The authors propose to restrict the performative effect of deploying model with parameter \theta to such a multiplier function \varphi_\theta, which is mainly a shift operator in the discussion of this paper and can be viewed as a strong restriction on shift pattern. Authors also show the benefit of scalability of performative gradient estimation based on this performative effect, and demonstrate the convexity of performative risk can be achieved through direction of performative effect instead of its magnitude, in the context of binary classification and shift operator. This finding is interesting and novel in the literature. Moreover, authors show the connection with robustness and regularization by a minimax reformulation.

**Strengths:**

1.	This paper clearly presents its assumption, main theorems with illustrative examples, which makes this paper easy to follow.
2.	This paper generalizes the performative shift pattern in former literature without restricting to location-scale family, and reveal another path to convexity of performative risk, which is innovative.
3.	Connection with robustness and regularization is also presented to strengthen the background of this paper.

**Weaknesses:**

1.	The main weakness is the generality of the performative effect, \Pi \theta, a shift on covariate, since it's the factual pattern considered in this paper. For general \varphi_\theta, this paper doesn't discuss how to identify or effectively estimate such transformation function \varphi_\theta. Moreover, the knowledge on shift matrix \Pi is also susceptible, since the specific mechanism of the distribution shift is usually unknown in the context of performative prediction.
2.	In the synthetic experiments, I think the authors should explore more abundant \varphi_\theta and show the benefit of scalability for high-dimensional \theta. In fact, the experiments mainly shown in the main paper is still restricted to the 2-dimensional simple setting of Izzo et al. 2022 and neither validate such scalability nor explore general \varphi_\theta. In addition, the colors used for denote different methods should be more distingushable.

**Questions:**

1.	Can the authors provide some intuitions on the dimension-freeness of \hat{G}_\theta^{R P}? I think it's a bit counterfactual since the shift \Pi \theta is put on the covariate X.

**Limitations:**

See weaknesses.

---

> ### Author Rebuttal · Authors · 2024-08-06
>
> We thank the reviewer for his/her positive review and address his/her questions below.
>
> **Knowledge of $\Pi$** We agree with the reviewer that knowing $\Pi$ entirely is indeed a limitation. In addition to the answers on this point given in response to reviewer PyL2 we point out that, from a theoretical point of view, the convexity arguments obtained in the paper require that $\Pi$ is known: even in the simpler cases, the performative risk would not be convex if it was to be optimized w.r.t. both $\theta$ and $\Pi$. From a more practical point of view, for linearly parameterized shifts, it is very natural to estimate $\Pi$ by ridge regression along the successive deployments of the model: the ridge penalty ensures that initially the estimate of $\Pi$ is close to zero making the RPPerfGD updates very similar to those of RGD and it is easy to check that order $d$ deployments are enough to obtain a non void estimate of $\Pi$. Of course, the behavior of the RPPerfGD algorithm that uses the values of $\Pi$ estimated along the trajectory of successive model deployments would need to be studied more precisely (probably requiring more specific choices of stepsizes than in the convex case). From a practical point of view however,  we provide in the global response PDF an additional figure pertaining to the example of Section 6 showing that in this case this approach is very effective. We will add this discussion in the final version of the paper.
>
> **Comments on the Experiment Section**
> We changed the colors to more contrasted ones. Note that figure 2b explores the impact of dimension with 7 dimensions, and the Housing dataset is also in 8 dimensions so our experiments are not limited to the 2-dimensional setting, and experiments in higher dimensions exhibits similar behaviors than in two dimensions. We also refer the reviewer to our answer to eHFF and our additionnal experiments, and would be happy to provide other similar plots during the discussion periods if a specific point can add value to the paper.
>
> **Dimension Freeness of the Covariance Estimator** The computations in Appendix A.1 shows that estimating the performative gradient is in this case fully equivalent, up to multiplication by $\Pi^T$,  to the estimation of the mean of a multivariate Gaussian with a $\sigma^2 I$ covariance matrix. Hence, the covariance matrix of the estimator is indeed dimension-independent. We agree with the reviewer however that in terms of interpretation of this finding there is a hidden dependence in the dimension here as the RMS error in estimating the mean of a multivariate normal is scaling as the square root of the dimension.

---

> > ### Comment · Reviewer_uddR · 2024-08-14
> >
> > Thanks for the response and clarification. I'll keep my evaluation unchanged.

---

> > > ### Author Response · Authors · 2024-08-14
> > >
> > > We thank the reviewer for his/her answer and feedback that contributed to improve our paper.

---

### Official Review · Reviewer_PyL2 · 2024-07-13

**Soundness:** 3
**Presentation:** 3
**Contribution:** 2
**Rating:** 5
**Confidence:** 2

**Summary:**

The authors address the challenge of performative prediction, a scenario in which the predictor's outcomes influence the underlying data distribution. They introduce a novel formulation for the gradient of the performative risk, thereby enabling the implementation of stochastic optimization methods. This new formulation offers an advantage over existing approaches by producing a gradient estimator with lower variance, particularly in cases where the data distribution shift is linear. Additionally, the authors establish a weaker sufficient condition for the convexity of the performative risk in situations involving linear classifiers and linear distributional shifts. Furthermore, they demonstrate that transform-invariant learning leads to parameter regularization. Empirical evaluations reveal that the proposed method achieves superior stable accuracy compared to existing techniques.

**Strengths:**

This well-written paper addresses a crucial problem in performative prediction that is highly relevant to the conference. The authors' theoretical contributions are clearly presented.

The proposed estimator for the performative gradient is novel and innovative. Its advantages over existing estimators are verified through an analysis of the estimator's variance under the linear shift scenario. The lower variance of this estimator results in faster convergence of the stochastic gradient descent, demonstrating its practical utility.

The authors' finding of a sufficient condition for the convexity of the performative loss under the linear shift setup is also a significant contribution. The benefits of this condition are adequately demonstrated, particularly in practical examples where existing techniques fail to verify the convexity of the performative loss.

The experimental results effectively demonstrate the superiority and stability of the proposed estimator compared to existing methods. These results indicate that the authors' method achieves greater stability while simultaneously improving accuracy, underscoring its practical applicability.

**Weaknesses:**

One potential limitation of the study is that the authors' analyses are primarily limited to linear shift cases. The linear shift assumption is relatively strong and may not hold in many practical scenarios. Moreover, the variance reduction effect is confirmed only under the linear shift setup, leaving the benefits of the proposed estimator in other situations unclear.

The implications of Theorem 4 require further clarification. The claim that the magnitude of $\theta^*$ becomes smaller for a larger $\Pi$ is questionable, as the norm $\\cdot\_\Pi$ is also affected by the magnitude of $\Pi$. This relationship warrants a more detailed explanation or additional analysis.

The assertion that knowing $\varphi$ is more practical than knowing $p_\theta$ may be an overstatement. Both assumptions appear to be equally unrealistic in practical applications, and this comparison could benefit from a more nuanced discussion.

**Questions:**

- How might your approach be extended to non-linear shift scenarios?

**Limitations:**

The authors adequately address the limitations and potential impacts.

---

> ### Author Rebuttal · Authors · 2024-08-06
>
> We thank the reviewer for his/her positive review and address his/her questions below.
>
> **Restriction to Linear Shifts** Theorem 2 proves convexity under general assumptions thanks to the linearity of the shift, and it is unlikely that convexity holds without this assumption. However, our approach to compute the performative gradient can also be used for non-linear shifts; it will only lack the theoretical guarantees. Thus, we agree with the reviewer that it is a limitation of our work, but we believe that it is an assumption that is more general than those found in previous works on performative learning (not being limited in particular to "small" performative effects). On the variance reduction effect, we also agree that we proved it for the specific combination of Gaussian noise and quadratic loss only. The proof given in appendix A.1, however, suggests that it could also be extended to non-linear drifts in the Gaussian/quadratic case. The comparison between both estimators is an open question that deserves a more systematic comparison, using other distribution/loss function combinations (e.g., Laplace noise could be more favorable for the use of the score function estimator as $\nabla_\theta \log p_\theta(z)$ would then have constant magnitude), and exceeds the scope of this paper.
>
> **Effect of Norm of $\Pi$** We agree with the reviewer that the comment under Theorem 4 is a bit hasty, we will make it clearer in the final version. The effect can be seen as follows: if $\Pi$ gets larger the r.h.s. decreases while the l.h.s. increases for identical values of $\theta$ (due to the fact that the norm depends on $\Pi$ as noted by the reviewer); hence values of $\theta$ have to be "smaller". The example given in lines 282--286 also illustrates this effect.
>
>
> **Knowledge of $\varphi$** We refer to the responses to reviewers eHFF and uddR on this point.
>
> **Questions on Extension to Non-Linear Shift Scenarios** As discussed above, this case is similar from a practical point of view and the same learning algorithm can be used but convexity of the performative learning would be most likely lost. Extension to transformations other than shifts however should be more difficult due to the necessity to consider the derivative of the log-Jacobian term. Note that the shift only needs to be linear in an embedding space, which covers already a large spectrum of situations.

---

> > ### Comment · Reviewer_PyL2 · 2024-08-14
> >
> > I appreciate the authors' response and maintain my positive assessment.

---

> > > ### Author Response · Authors · 2024-08-14
> > >
> > > We thank the reviewer for his/her positive feedback confirming that our rebuttal addressed his/her concerns. We also thank the reviewer for his/her feedback that will improve the final version of the paper.

---

### Author Rebuttal · Authors · 2024-08-06

We thank the reviewers for their careful assessment of the manuscript and their helpful suggestions to improve the clarity. We are happy to see that reviewers recognize that our "proposed estimator for the performative gradient is novel and innovative" (PyL2) and that our assumptions are "flexible and general" (eHFF) and the connection with robustness "strengthen the background of this paper" (uddR). We provide in PDF a new experiment with estimation of $\Pi$ during the successive deployments, reporting the impact on the accuracy and the convergence of $\Pi$. More details and the code of this experiment will be included in the final version of the paper. We answer specifically to each reviewer below.

---

### Decision · Program_Chairs · 2024-09-25

**Decision:**

Accept (poster)

**Comment:**

This paper proposed a new method to estimate the gradient under a performative distribution shift setting. They relied on a specific "Push-forward Performative Model" assumption. While some reviewers pointed out that the experiments are relatively of a small scale, and the estimation of the pushforward map is not easy, all reviewers agreed that the paper should be accepted due to the innovations presented; especially due to the theoretical convexity and variance-reduction results.